# How Medical Staff Alleviates Job Burnout through Sports Involvement: The Mediating Roles of Health Anxiety and Self-Efficacy

**DOI:** 10.3390/ijerph191811181

**Published:** 2022-09-06

**Authors:** Xiuyu Chen, Longjun Jing, Huilin Wang, Jingyu Yang

**Affiliations:** 1School of Physical Education, Hunan University of Science and Technology, Xiangtan 411201, China; 2China Athletics College, Beijing Sport University, Beijing 100084, China; 3School of Business, Hunan University of Science and Technology, Xiangtan 411201, China; 4Department of Medical Bioinformatics, University of Göttingen, 37077 Göttingen, Germany

**Keywords:** medical staff, job burnout, sports involvement, health anxiety, self-efficacy

## Abstract

In the current healthcare environment, job burnout among medical staff is increasingly evident. Burnout not only affects the mental and physical health and career development of individuals but also affects the quality of care and the doctor–patient relationship. This paper investigates the influence of sports involvement on burnout in medical staff based on the job demands–resources theory, focusing on the mediating role of health anxiety and self-efficacy in the relationship between sports involvement and job burnout. A questionnaire survey was used to collect data from 444 medical staff in public hospitals in Wuhan, China. Structural equation modeling (SEM) with a bootstrapping approach was conducted to test the hypothesis and mediating effects. It was found that health anxiety and self-efficacy played a significant mediating role between sports involvement and job burnout. The results indicate the important role that sports involvement plays in addressing burnout, revealing that decreasing health anxiety and increasing self-efficacy attenuated job burnout. This finding suggests that hospital administrators should not only pay attention to medical staff’s health conditions and improve their enthusiasm for work but also encourage them to become more engaged in sports.

## 1. Introduction

During the past two decades, the world has experienced outbreaks of several highly infectious diseases and serious public health emergencies. Expecting to experience increases in workload, medical staff also face problems such as conflicts between professional obligations and personal safety, coupled with high stress, infrequent interpersonal contact, irregular work, and not enough time to rest. Long-term exposure to these stressful situations can have certain effects on the physiological, psychological, and social health of medical staff, leading to hypertension, anxiety and depression; this, in turn, affects work efficiency, which may eventually lead to job burnout [1,2]. After the COVID-19 pandemic broke out in 2020, the mental health of medical staff received the attention of health administration departments, medical institutions, and society, and various research studies and interventions have been undertaken. Currently, China is in the midst of a regular epidemic, and understanding the health status of medical staff can help in the development of management methods and the implementation of interventions. This paper investigated the mental health status of medical staff in Wuhan, aiming to provide a theoretical basis for hospitals to develop health strategies for medical staff in a targeted manner.

Health anxiety is a condition characterized by excessive preoccupation with an individual’s health [3], defined as a state consisting of psychological and physical symptoms brought about by a sense of apprehension about a perceived health threat. The COVID-19 pandemic has introduced additional elements of fatigue, strain, stress, loss, and grief for healthcare workers [4]. Many healthcare workers experienced increased workloads in the face of short staffing and shortages in critical personal protective equipment. Furthermore, many healthcare workers place the well-being of others before themselves. However, it can ultimately be harmful if it delays or prevents workers from getting the help that they need for their health and well-being. In recent studies, the prevalence of health anxiety among medical staff ranged from 30.1–72.3%, much higher than that of the general population [5]. Health anxiety and high workload can lead to psychological distress and job burnout among medical staff [6]. Medical staff on the front lines of the epidemic play an important role in controlling the epidemic, and the psychological distress of health anxiety can lead to physiological disorders, which further affects the quality of healthcare services. Therefore, ensuring the psychological health of medical staff can also ensure a certain degree of public health. In addition, health anxiety as a physiological problem is the cause of job burnout in medical staff. Stress, stress-related disorders, such as depression, anxiety, perceived impaired memory, diabetes, and metabolic syndrome [7] and some health impairments, such as headaches, sleep disturbances, neck and back pain, musculoskeletal diseases, and cardiovascular diseases [8], are associated with job burnout. Pre-existing studies and systematic reviews on health anxiety show that the factors influencing medical staff’s health anxiety are mainly reflected in demographical variables such as age, gender, education level, and occupation [9,10]. Although scholars have conducted more extensive research in the area of psychological symptoms of anxiety, there is limited research on the relationship between health anxiety and job burnout. Therefore, the objectives of this study are as follows: (1) to understand the health anxiety of medical staff under the public health emergency; (2) to explore physiological symptoms that influence the job burnout of medical staff; and (3) to suggest strategies to decrease health anxiety and alleviate burnout symptoms of medical staff.

Self-efficacy refers to the subjective prediction of one’s ability to accomplish a certain task, as well as the individual’s tendentious judgment and feeling on whether one’s behavior can achieve a certain goal [11]. It has three implications: first, it belongs to the category of perception ability, but it is not equal to ability; second, the expectation of achieving a certain goal comes before the activity; third, it is a subjective judgment on whether one can achieve a certain goal. Different from previous studies that use positive psychological factors (i.e., self-efficacy) as the main factor of job burnout, negative factors (i.e., health anxiety) are used as mediating variables to influence the extended model of the job demands–resources (JD-R) theory. Previous studies have suggested that the relationship between physiological factors and job burnout is underexplored. However, a study by von Känel et al. found the influence of physiological factors (i.e., sports involvement) on job burnout based on the JD-R theory and further explained and predicted the occurrence of job burnout [12]. Sports involvement may be indirectly associated with job burnout, mediated by self-efficacy and health anxiety. Understanding the job burnout of medical staff will help the development of related theories and help this group improve physiological and psychological health.

The following sections are structured as follows: Section 2 reviews the literature related to JD-R theory and presents hypotheses and conceptual models. Section 3 introduces the participants, data collection methods, questionnaire composition, and data analysis methods. Section 4 describes the results of the data analysis and tests the hypotheses. Section 5 discusses the results and proposes the particular implications. Section 6 summarizes the central ideas of the paper and discusses prospects for further research.

## 2. Literature Review and Hypothesis Development

### 2.1. Job Demands-Resources Theory

Demerouti and Bakker et al. [13] proposed the job demand–resources model (JD-R) in 2001. The JD-R model consists of two core assumptions. The first is that any occupation contains two stress-related risk factors: job demands and job resources. Job resources are the physical and psychological factors that are associated with the energizing process, while job demands are the physical or emotional stressors that are associated with the health impairment process. The model states that when job demands are high and job resources/positives are low, burnout increases. More specifically, job resources motivate individuals’ work engagement, while lack of sufficient job resources prevents goal accomplishment, which causes a sense of failure and frustration. High job demands may lead to depletion of energy and further endanger the individual’s health [14]. Although there is a consensus in most studies that the combination of high job demands and low job resources is an important explanation for burnout [15], most cases that lead to burnout still cannot be explained. Burnout is not only determined by environmental factors but is also influenced by personal factors and the interaction between people and the environment [16]. Previous papers have focused on opportunities or resource factors such as social support, job autonomy, task feedback, or self-efficacy to reduce the development of burnout [13,17], while the inhibitory effect of negative situations on burnout has been neglected. In recent years, scholars have called for a focus on the negative factors of burnout, which have a stronger impact on individuals than positive factors due to the “positive–negative asymmetry” [18]. Furthermore, too much focus on the positive antecedents of burnout mitigation is detrimental to the real situation individuals face in organizations. Much evidence shows that medical staff experience negative situations such as psychological problems in organizations, with health anxiety being the most harmful [19]. The data show that the detection rate of depression among medical staff in the context of the COVID-19 epidemic is as high as 67.4%, and the percentage of those with severe health anxiety symptoms is 45% [20]. A review of the literature revealed that there are few studies on the relationship between health anxiety and job burnout among medical staff. Thus, this paper investigates the mechanism of health anxiety and self-efficacy that influences job burnout.

### 2.2. Stress, Health Anxiety, Job Burnout, Sports Involvement, and Self-Efficacy

Under the normal of epidemic prevention and control, the medical industry is an important guarantee to avoid infection and ensure people’s lives and health, and the medical staff is constantly exposed to high stress due to the high risk of infection, isolation, caring for critically ill patients, and overwork. This difficult situation leads to mental health problems such as anxiety, fear, stress, insomnia, and depression [21]. The cross-sectional research method was adopted to collect data through a questionnaire survey of 404 Nepali healthcare workers. The research by Pandey et al. [22] found that the symptoms of stress, anxiety, and depression among healthcare workers were 28.9%, 35.6%, and 17.0%, respectively. Deng et al. [23] carried out a comparative analysis of 34 articles and verified that the prevalence of depression and anxiety was higher among healthcare workers than among the general public in China. Health anxiety is often found in combination with generalized anxiety, and obsessive-compulsive disorder [24], and the prevalence of it in the medical staff was higher than in the general population. One important factor leading to health anxiety in medical staff is the high rate of infection and mortality among them [25]. Medical staff caring for patients experience mental stress and physical fatigue. Phycological health problems such as sleep disorders [26], depression [27], sedentarism, obesity [28], and musculoskeletal pain [10] are closely linked to job burnout [29]. Job burnout is a state of work-related psychological exhaustion, and an increased risk for somatic diseases was highly associated with the exhaustion dimensions of job burnout [12]. Sports involvement, as part of disease prevention and treatment strategy, has been used in a variety of chronic conditions. Sports involvement decreases disease risk and chronic illnesses, such as cardiovascular disease, type 2 diabetes, and obesity, and can significantly improve pain and related symptoms with appropriate frequency, duration, and intensity [30].

People with higher self-efficacy can complete the original sports involvement plan when they confront difficulties [31]; self-efficacy could not only stimulate the motivation level of individuals but also determine the level of individual involvement in sports [32]. Research results show that people with disabilities who play sports have more pronounced self-efficacy compared to those who do not play sports [33]. Furthermore, physiological or emotional states are one of four sources of self-efficacy and can be used to increase self-efficacy [34]. Past research supports the notion that sports involvement has a positive effect on psychological well-being and physiological health. In particular, sports involvement reduces self-reported anxiety and depression [35] and serves to increase positive mood [36]. Therefore, sports involvement can effectively alleviate and control individuals’ negative emotions and help them improve their mental resilience and maintain a positive attitude [15]. Self-efficacy, as a positive self-belief perception, predicts job satisfaction and job performance and plays an important role in individuals [37]. Self-efficacy can influence negative emotions such as anxiety and depression and has a positive impact on an individual’s mental health and well-being [38], and in terms of engagement with work, it also influences job satisfaction, productivity, and positive aspiration [39]. The alleviation of emotional exhaustion and the enhancement of personal fulfillment are conducive to increasing individuals’ job satisfaction and reducing burnout. High self-efficacy has been found to reduce job burnout, such as work engagement and emotion regulation. Based on the above, these research hypotheses are proposed:

**Hypothesis** **1** **(H1).***Sports involvement is negatively associated with health anxiety*.

**Hypothesis** **2** **(H2).***Health anxiety is positively associated with job burnout*.

**Hypothesis** **3** **(H3).***Health anxiety mediates a relationship between sports involvement and job burnout of medical staff*.

**Hypothesis** **4** **(H4).***Sports involvement is positively associated with self-efficacy*.

**Hypothesis** **5** **(H5).***Self-efficacy is negatively associated with job burnout*.

**Hypothesis** **6** **(H6).***Self-efficacy mediates a relationship between sports involvement and job burnout of medical staff*.

### 2.3. Research Model

Considering these arguments on the relationship between sports involvement, health anxiety, self-efficacy, and job burnout, the conceptual model of the present paper to examine the research hypotheses is presented in Figure 1.

## 3. Methods

### 3.1. Participants

The researchers adopted the snowball sampling method, a sample is the participants you select from a target population to make generalizations about. The researchers selected medical staff from a three-tier public hospital in Wuhan. With the support of schools and hospitals, questionnaires were distributed by heads of departments to medical staff in their departments. The questionnaire was conducted from April to May 2022. Before the date of collection, the researchers conducted a pilot test with a sample of 50 medical staff from a hospital to assess the validity of our variables and the framing of questions. All participants voluntarily participated in the survey and earned a gift worth about 5 USD as a response incentive. The responses by participants were anonymous. Out of 500 distributed questionnaires, 56 uncompleted and inaccurate questionnaires were excluded, providing 444 valid responses, with an 88.8% effective response rate.

Table 1 lists the demographic characteristics of the residents who participated in the survey. Of them, (1) on gender, the ratio of male and female was unequal (male = 35.4%, female = 64.4%); (2) 49.8% were aged 31–39 years, and 41.0% were aged 40–49 years; (3) the majority of respondents surveyed had a college/university degree or higher (61.9%); (4) 70.3% of the respondents were nurses; and (5) the dominant percentage of the residents had more than six service years. The results of this survey are pretty close to the population statistics of medical staff, among the 444 medical and nursing staff, the age range was 24 to 72, and the average age was 34.15 ± 8.56.

### 3.2. Instrument and Measures

To measure sports involvement, three items were extracted from the research of Beaton et al. [40]. The scale exhibits good factor reliability and has been used in the sample of residents of a city [41]. The researchers measured health anxiety using four items from the Short Health Anxiety Inventory (SHAI) scale of Abramowitz, Deacon et al. [42]. Self-efficacy was measured using three items of the scale derived from Schwarzer and Jerusalem et al. [43]. The Maslach Burnout Inventory for General Survey (MBI-GS) developed by Schutte et al. [44] assessed burnout. The MBI-GS has been widely used among medical staff in China and has satisfactory reliability and validity [45]. All constructs were measured using a five-point Likert scale, in which the responses ranged from 1 (i.e., completely disagree) to 5 (i.e., completely agree). Questions relating to the respondents’ demographic characteristics, involving gender, age, educational background, profession, and length of service were also included in the questionnaire.

### 3.3. Statistical Analyses

Structural equation modeling (SEM) with AMOS 26.0 (IBM, Chicago, IL, USA) was used to examine the hypothesized model [30]. The researchers constructed a structural equation model of sports involvement and job burnout in medical staff and used path analysis and diagrams to explain the internal relationships among variables to verify our hypotheses. Additionally, the studies analyzed and verified the mediating effect of medical health anxiety and self-efficacy between sports involvement and job burnout. Finally, this study used the Bootstrap method proposed by MacKinnon et al. [46] to test the mediating effect of health anxiety and self-efficacy with a sample size of 5000 and a 95% confidence interval.

## 4. Results

### 4.1. Assessment of the Measurement Model Reliability and Validity

Cronbach’s α coefficient (Cα) and the composite reliability (CR) coefficient of the latent variables, as an acceptable approach proposed by Fornell and Larcker [47], were calculated for examination of reliability and discriminant validity. Table 2 exhibits the details of factor loadings (Loadings), Cronbach’s α coefficients, average variance extracted (AVE), and CR. First, Cronbach’s α coefficients of the variables were in the range of 0.726–0.865. Second, all the CR values were higher than 0.7, as suggested by Hair et al. [48]. Third, the AVE of all variables was between 0.472 to 0.863, implying an acceptable convergent validity. Therefore, all variables have acceptable convergent validity.

Pearson’s correlation coefficient depicts the extent that a change in one variable affects another variable. This relationship is measured by calculating the slope of the variables’ linear regression. Researchers verify the discriminative validity of the data by comparing the correlation coefficient of each variable with the square root of the AVE. Table 3 illustrates the mean and standard deviations (SD) of sports involvement (SPI), health anxiety (HEA), self-efficacy (SEE), and job burnout (JOB). Results show that all the variables have good discriminant validity, as all correlation coefficients were less than the square root of the AVE. Additionally, for examining the extent to which common method variance (CMV) is present in the data, Harmon’s single factor test approach was used [49]. In a single factor test, one examines the unrotated factor solution to determine the number of factors that are necessary to account for the variance in the variables. The maximum factorial variance explained was 30.13%, which was less than the critical standard of 50%; thus, there was no common method bias in this study.

### 4.2. Hypothesis Testing Results

The structural modeling results indicated that the structured model has a good fit: the chi-square of degrees of freedom (χ^2^/df) = 2.146, the goodness of fit (GFI) = 0.951, the normed fit index (NFI) = 0.942, the comparative fit index (CFI) = 0.968, the Tucker–Lewis index (TLI) = 0.959, and the root mean square error of approximation (RMSEA) = 0.051, with reference to the suggested value of Joseph F. Hair et al. [48]. Table 3 lists the mean, standard deviation, and correlation among the variable. Significant correlations were found between the independent variables, the mediators, and the dependent variables, which provided preliminary support for the verification of six hypotheses. The structural path model results are presented in Figure 2. Sports involvement was significantly related to health anxiety (*β* = −0.234, *p* < 0.001) and self-efficacy (*β* = 0.592, *p* < 0.001), supporting H1 and H4. The effect of health anxiety on job burnout was statistically significant (*β* = 0.380, *p* < 0.001), supporting H2. The effect of self-efficacy on job burnout was statistically significant (*β* = −0.149, *p* < 0.05), supporting H5.

### 4.3. Mediating Effects

This study followed the recommendation of Bollen and Stine [50] and used the bootstrapping approach to verify the mediating effects. The results of 5000 bootstrap samples, with a 95% confidential interval, are presented in Table 4; all Z values were greater than 1.96, and there was no zero value in the 95% confidential interval. Moreover, it showed that mediation occurred between sports involvement and job burnout through health anxiety (standardized indirect effect = −0.071, *p* < 0.001), which provides support to H3. It also showed that mediation occurred between sports involvement and job burnout through self-efficacy (standardized indirect effect = −0.075, *p* < 0.001), which provides support to H6. The findings indicate that medical staff who have more sports involvement, higher self-efficacy, and less health anxiety are less likely to develop job burnout, and the two mediating variables of health anxiety and self-efficacy explained 18.3% of the variance in job burnout.

## 5. Discussion

### 5.1. Contribution

This study has made the following contribution to research on job burnout. First, it clarified the positive impact of sports involvement on alleviating individual job burnout by investigating burnout and sports involvement among 444 medical staff and further explored the specific mechanisms of the impact of sports involvement on job burnout by adding two psychological variables—health anxiety and self-efficacy—as mediating variables. The results indicated that the positive effect of sports involvement on job burnout was fully mediated by health anxiety and self-efficacy, and the two mediating variables of health anxiety and self-efficacy explained 18.3% of the variance in job burnout. According to previous studies, job burnout has been defined as a long-term reflection of emotions and stress at work and a negative work-related state of mind as a comprehensive indirect reflection of physical, mental, and emotional aspects. From this perspective, both emotional exhaustion and negative evaluations of oneself can affect an individual’s attitude toward work. Health anxiety is an excessive focus on the fear of illness or a strong belief in illness, and some people suffering from health anxiety will experience some physical symptoms of chest pain, abdominal pain, facial pain, or headaches. Essentially, it exerts an influence on job burnout by changing the individual’s emotional exhaustion of burnout. Studies have been conducted linking excessive physical fatigue to emotional exhaustion. Self-efficacy, however, influenced job burnout by changing one’s positive self-perceptions. From previous studies, self-efficacy was found to significantly and negatively predict the level of job burnout. The higher the level of self-efficacy for medical staff, the lower the level of job burnout in the same professional environment and the more they believe they can handle occupational stress and have the ability to perform their jobs when faced with stress and professional dilemmas. In contrast, medical staff with low self-efficacy find it difficult to adjust their state from occupational distress and tend to consume more energy and approach their work with a negative attitude, which eventually aggravates the degree of job burnout and triggers a series of physical and mental health problems.

Second, this paper is the first attempt to explain the effects of sports involvement on health anxiety, self-efficacy, and job burnout using the JD-R theory. JD-R theory explores the effects of job characteristics on job burnout in terms of job demands and job resources, and previous studies have focused on opportunity or resources factors at the physical, social, and psychological levels [13,51]; in contrast, negative factors are the focus of this study [16]. Therefore, this paper incorporates negative factors (i.e., health anxiety) and opportunity factors (i.e., self-efficacy) into the JD-R model to further enrich and expand the model and enhance its explanatory power. In addition, a series of studies have shown that sports involvement has a positive impact on the promotion of physical and mental health of medical staff and that exercise intervention therapy has been applied in the treatment of job burnout, but the reasons for the impact of sports involvement on job burnout remain unexplained at present. However, the effects of sports involvement on improving the prognosis of cardiovascular, skeletal, and joint diseases, improving sleep quality, and reducing anxiety and depression levels have been verified [27]. The theoretical presupposition of this study using the JD-R model to explain the relationship between sports involvement, health anxiety, self-efficacy, and job burnout is valid.

### 5.2. Practical Implications

It has been shown that the high-pressure and high-load working environment, and the high-risk occupational characteristics of medical staff consume their energy, resulting in a high incidence of job burnout. Long-term burnout not only depletes the physical and mental health of medical staff but also reduces the quality of the healthcare services provided and leads to conflicts between doctors and patients [52]. This study confirms the mechanism of the effect of sports involvement on job burnout in medical staff; in addition, we should also see the contribution of sports involvement to the individual’s physiological and psychological aspects. During this COVID-19 pandemic, the medical staff of Wuhan’s facing shelter hospital led patients to exercise collectively, using square dance, Tai Chi, and push-ups to help their physical recovery. In addition, empirical studies have shown that exercise is effective at improving the quality of life of medical staff [53]. Even facing heavy workloads, medical staff can exercise to regulate their ongoing phycological and physical exhaustion. Although the public increasingly recognizes the importance of sports involvement as an effective means of enhancing physical and mental health, the high intensity and stressful nature of medical work are such that the prospects of health workers engaging in exercise are bleak. The reasons for this include the following. First, medical staff has not developed the habit of exercise or the concept of lifelong sports. Second, the shortage of venues and facilities and the lack of guidance affect the sports involvement of medical staff. Third, due to the special nature of healthcare work, medical staff are more likely to suffer from physical and mental fatigue, which directly affects their enthusiasm and attitudes toward exercise.

Considering the impact of sports involvement on health anxiety, self-efficacy, and job burnout, this study makes the following suggestions for medical staff. First, more attention should be paid to strengthening medical staff awareness of the concept of lifelong sports. The government and hospital managers should incorporate sports exercise venues for medical staff into the overall planning of hospital development, regularly hold various lectures on sports topics, propagate the skills and methods of scientific exercise, and carry out more sports activities, such as jumping rope, table tennis, and other competitive projects, to guide medical staff to actively exercise. Second, more scientific exercise programs should be pursued. Hospitals should rely on sports managers to customize personal sports prescriptions for medical staff to provide guidance on when to exercise and to have appropriate knowledge and information about sports at the same time as exercise.

## 6. Conclusions

In response to the proposed research objectives, this study is motivated by the widespread phenomenon of burnout among medical staff in China. Medical staff is faced with a heavy workload, tedious work tasks, and long working hours, and their common responsibilities involve the assessment, planning, and administration of daily patient treatment and health management. They are more likely to suffer from health disorders and experience symptoms such as neck and back pain, fatigue, and sleep disturbance that may cause anxiety and depression, leading to burnout. Moreover, the result demonstrates that health anxiety and self-efficacy are important factors that affect medical staff sports involvement and job burnout. In particular, sports involvement affects job burnout indirectly through the mediating effects of health anxiety and self-efficacy. Therefore, this study recommended that hospital managers pay attention to medical staff’s health conditions, improve their work motivation and encourage them to be more engaged in sports.

This study has certain limitations. First, the study design was cross-sectional. Thus, the relationship between sports involvement, health anxiety, self-efficacy, and job burnout cannot be assumed to be causal, and longitudinal research is necessary to examine the causal relationships in this study for the future. Second, we adopted the snowball sampling method; thus, the reported information may not accurately reflect the underlying values of each variable, unlike random sampling. Third, the two mediating variables of health anxiety and self-efficacy explained 18.3% of the variance in job burnout, but there remains room for a lot of unexplained variation. Further research can provide more possible variables based on this.

## Figures and Tables

**Figure 1 ijerph-19-11181-f001:**
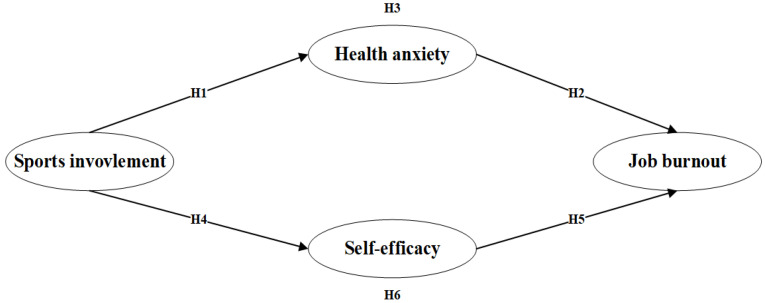
The hypothesized model.

**Figure 2 ijerph-19-11181-f002:**
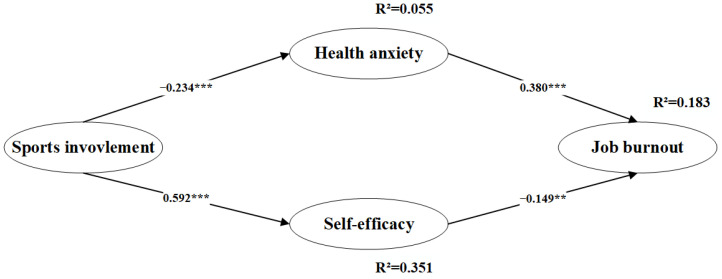
Structural equation modeling of the hypothesized model. ** *p* < 0.01, *** *p* < 0.001.

**Table 1 ijerph-19-11181-t001:** Demographic profile of the survey participants (N = 444).

Profiles	Number of Respondents	Proportion
Respondent gender (%)		
Male	157	35.4%
Female	287	64.6%
Respondent age (%)		
≤25	5	1.1%
26–30	134	30.1%
31–39	221	49.8%
40–49	186	41.9%
≥50	32	7.2%
Respondent education level (%)		
Vocational school	169	38.0%
College/University	191	43.0%
Master or Ph.D.	84	18.9%
Respondent profession (%)		
Doctor	132	29.7%
Nurse	312	70.3%
Length of service (%)		
Below 5 years	89	20.0%
6–10 years	308	69.4%
More than 11 years	47	10.6%

**Table 2 ijerph-19-11181-t002:** Reliability and validity test.

Dimensions	Loadings	Cα	AVE	CR
Sport Involvement		0.858	0.683	0.865
SPI1: Exercise plays a central role in my life.	0.773			
SPI2: I enjoy discussing exercise with my friends and family.	0.917			
SPI3: I enjoy exercise.	0.782			
Health Anxiety		0.797	0.498	0.798
HEA1: Fear of having a serious illness.	0.625			
HEA2: Ability to take the mind off health throughs.	0.744			
HEA3: Feeling at risk for developing the illness.	0.727			
HEA4: Ability to enjoy life if have an illness.	0.723			
Self-Efficacy		0.724	0.472	0.726
SEE1: I can always manage to solve difficult problems if I try hard enough.	0.624			
SEE2: It is easy for me to stick to my aims and accomplish my goals.	0.644			
SEE3: I can still be calm when facing difficulties because I can rely on my coping abilities.	0.783			
Job Burnout		0.862	0.616	0.864
JOB1: I feel fatigued when I get up in the morning and must face another day on the job.	0.783			
JOB2: I feel burned out from my work.	0.837			
JOB3: I can’t easily create a relaxed atmosphere with my recipients.	0.820			
JOB4: I can’t feel exhilarated after working closely with my recipients.	0.691			

**Table 3 ijerph-19-11181-t003:** Discriminant validity test.

Construct	Mean	SD	SPI	HEA	SEE	JOB
SPI	3.330	1.050	**(0.826)**			
HEA	3.049	0.838	−0.187 **	**(0.706)**		
SEE	3.560	0.674	0.493 **	−0.148 **	**(0.687)**	
JOB	2.818	0.796	−0.148 **	0.348 **	−0.177 **	**(0.785)**

The square root of the average various extracted (AVE) is in diagonals (bold); off diagonals are a Person’s corrections coefficients. ** *p* < 0.01.

**Table 4 ijerph-19-11181-t004:** Standardized direct, indirect, and total effects.

	Point Estimate	Product of Coefficients		Bootstrapping				Two-Tailed Significance
				Percentile 95% CI		Bias-Corrected 95% CI		
		SE	Z	Lower	Upper	Lower	Upper	
Direct Effects								
SPI → HEA	−0.234	0.066	−3.493	−0.362	−0.102	−0.359	−0.099	0.000 (***)
SPI → SEE	0.592	0.052	11.170	0.485	0.689	0.482	0.687	0.000 (***)
HEA → JOB	0.380	0.066	5.938	0.245	0.496	0.253	0.502	0.000 (***)
SEE → JOB	−0.149	0.074	−2.014	−0.298	−0.009	−0.292	−0.002	0.044 (*)
Indirect Effects								
SPI → JOB	−0.177	0.053	−3.340	−0.285	−0.077	−0.284	−0.076	0.000 (***)
Total Effects								
SPI → HEA	−0.305	0.064	−4.765	−0.348	−0.097	−0.345	−0.095	0.000 (***)
SPI → SEE	0.667	0.053	12.584	0.481	0.688	0.480	0.687	0.000 (***)
HEA → JOB	0.451	0.063	7.159	0.257	0.506	0.260	0.508	0.000 (***)
SEE → JOB	−0.224	0.103	−2.175	−0.392	−0.011	−0.386	−0.017	0.028 (*)

Standardized estimations of 5000 bootstrap samples. * *p* < 0.05, *** *p* < 0.001.

## Data Availability

Not applicable.

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
