# Peer review of "How Medical Staff Alleviates Job Burnout through Sports Involvement: The Mediating Roles of Health Anxiety and Self-Efficacy"

_ijerph, 2022, doi:10.3390/ijerph191811181_

Round 1

Reviewer 1 Report

I read the paper with interest as it addresses a persistent issue of job burnout. As an enthusiastic individual regularly engaging in sports, I also found personal interest in the results of the study. However, there are several critical issues with this paper that do not allow me to recommend an accept for this clearly and fairly well-written paper.

First, I identified issues with measurement items for sports involvement (what a disappointment -- i dislike running despite being regularly involved in sports!) and health anxiety. Next, there are issues with instrument validation. Finally, there are issues with interpretation of results and mediation analyses conducted. All these critical issues should be thoroughly addressed in a major revision. Note that the discussion should be reworked accordingly.

Detailed comments:

- Although the authors outline the problem "The psychological factor of health anxiety and self-efficacy can only partly explain burnout, and other factors that contribute to burnout seem to be physiological factors and other factors", this study does not address this issue since job burnout is directly associated only with health anxiety and self-efficacy.

- Statements like "Through sports involvement, the psychological factors of self-efficacy and health anxiety indirectly affect job burnout." are inconsistent with the research model in which the opposite is suggested (i.e., sports involvement is indirectly associated with job burnout, mediated by self-efficacy and health anxiety).

- The statement "healthcare industry is one of the most dangerous work environments" needs to be substantiated with adequate references. I come from a family of physicians and I simply don't see this "dangerous work environment".

- Hypotheses need to be rephrased. A cross-sectional research design allows to determine only associations/relationships between theoretical constructs (and not effects/impact of one construct on another). For example:

"Hypothesis 1 (H1). Sports involvement has a positive impact on health anxiety."

==> "Hypothesis 1 (H1). Sports involvement is positively associated with health anxiety."

Note that this needs to be addressed thorough the whole paper.

- H1 -- Since sports involvement reduces health risks, I expected this hypothesis to go the other way around (i.e., a negative association with health anxiety). H2 -- A similar issue with health anxiety-job burnout, I expected a positive association (i.e., the more anxious, the more burnt out). If H1 and H2 are not mixed up, I would expect a convincing argument for supporting these counter-intuitive hypotheses. Note that the results support a mix-up.

- H3 -- It is unclear what a "negative mediating effect" is supposed to be. A construct can either mediate a relationship between two constructs or not. Similar question for H6.

- Figure 1 -- What is the meaning of the dashed line?

- Section 3 -- It seems that some instructions were left at the beginning of this section.

- Section 3.1 -- How were participants approached? The paper indicates them as "medical staff in Wuhan" but how were they recruited? Were they part of a single, multiple or all healthcare institutions? Which types of healthcare institutions were they from -- only hospitals? Which tier(s)?

- Measures -- There is a mismatch between the theoretical construct sports involvement and items which focus only on running. The theoretical construct needs to be renamed and operationalized differently since there are obviously several other sports besides running.

- Health anxiety -- Items seem quite poor as it is unclear what kind of illness is suggested. There are significant differences between different types of diseases in terms of their probability and severity, and focusing on a single non-defined illness suggests that there is a high probability that the respondents did not provide meaningful answers here.

- Instrument validation -- AVE for health anxiety and self-efficacy is below the standard threshold .50, and CR is not above .80 for these constructs. This indicates issues with convergent validity and contradicts the authors' claims, such as "Therefore, all variables have high convergent validity." Excluding certain items might solve these issues.

- Typo in Table 3: "MD" ==> "SD"

- "Significant correlations were found between the independent various, the mediators, and the dependent various, which provided preliminary support for the verification of six hypotheses."

==> Typos "various" should be "variables"
==> Correlations tell us nothing about H3 and H6

- Figure 2 -- It includes a path with ** significance.

- R² for health anxiety is too low (.055) to establish an association with sports involvement. Although there is a significant path, the variance explained is way too low to consider it as a meaningful association. Therefore, I don't see support for H1 or H3 in these results.

- I am not familiar with Bollen & Stine (1990) but I am familiar with testing for mediation. Typically, indirect effects for each mediation should be tested separately. For example, H3 should be tested for indirect effects from sports involvement to job burnout through health anxiety. If the indirect effect is significant, we have mediation. The approach taken in the paper however reports on the overall indirect effects -- we probably have mediation but it remains unclear whether H3, H6, both or none of them are supported.

- Table 4 -- It includes a * significance, but not a ** one.

- Some relevant references from 2022 may be missing.

Author Response

Dear reviewer,
We sincerely thank you for examining our manuscript and providing helpful comments to guide our revision. We have tried our best to revise the manuscript according to your kind and construction comments and suggestions. Please find the following detailed responses to your comments and suggestions. We would like to thank the referee again for taking the time to review our manuscript.

Reviewer 2 Report

This is an interesting manuscript that  aims at studying the role of sports for burnout through decrease of health anxiety and increase of self-efficacy. I have plenty of comments because I believe that the MS can be made more clear. Good luck!

General comments:

According to the title, the article aims at studying the role of sports for burnout through decrease of health anxiety and increase of self-efficacy. This is especially well demonstrated in Figure 1. The authors use structural equation model (SEM) for analysis.

The introduction does not clearly lead the reader into the study. For example, on Page 2 (1st whole paragraph) the authors state:

“Studies have shown that stress is associated with the level of health anxiety and prolonged exposure to a high-pressure work environment that triggers higher levels of health anxiety in individuals”

and

“Health anxiety and high workload can lead to psychological distress and job burnout among medical staff”

-        Here it is unclear how health anxiety and exposure in high-pressure work environment/high work load affects stress and how they can be separated.

Further, the authors state:

“Although scholars have conducted more extensive research in the area of health anxiety, there is very limited research on the relationship between health anxiety and job burnout” 

-        If you do not separate health anxiety and work load / work environment, you cannot get knowledge about health anxiety.

Further unclarities: one aim is to investigate health anxiety and risk factors for burnout (what are the ‘risk factors’ here?); later you create strategies for improving job satisfaction (how do you define job satisfaction?). Why not create strategies for decreasing health anxiety?

On page 6 the authors state

Additionally, the studies analyzed and verified the mediating effect of medical health anxiety and self-efficacy between sports involvement and job burnout.”

and on pages 5-6 they define that they used Short Health Anxiety Inventory scale, self-efficacy scale, sports involvement scale and burnout inventory. Here one can finally understand what actually is measured and what is not.

Job resources-demands (JR-D) theory is also used but the concepts are not specified clearly enough.

Taken together, the authors need to reorganize the introduction and elaborate the aims. For clarity, I suggest concentrating on health anxiety, self-efficacy and burnout, and defining them in the introduction. JR-D theory and it’s connection with the above mentioned concepts should also be clearly introduced either in the introduction or in chapter 2. I might suggest skipping job satisfaction, work load and work environment from the introduction and elaborate the relationship between them and the three main concepts (health anxiety, self-efficacy and burnout) in chapter 2. This is especially important since you use SEM; for using SEM one needs to theorize the structure of the phenomenon very clearly. Further, you should be able to show (in Chapter 2) how the main concepts of your study can be separated from all other concepts.

Statistics

You use structural equation model (SEM) for analysis. It is not very well defined, especially since SEM usually can include various mathematical methods. You mention some methods (Harmon’s [should be Harman’s] single factor test, structural path model), but the text is extremely short.

For me it is a bit difficult to understand (without you explaining the reasons), why you used Harman’s single factor test, since it is a method for exploratory factor analysis, while in SEM confirmatory factor analysis is usually used. So, please explain.

Details:

Page 1, bottom and Page 5:

“This paper investigated the mental health status of medical staff in Wuhan, aiming to provide…”

“The researchers selected medical staff in Wuhan as research objects, considering that it is Wuhan was the area where the COVID-19 pandemic broke out, the related measures implemented in Wuhan, such as the lockdown, and the traffic restrictions implemented to reduce the spread and fatality of the disease.”

-        The pandemic broke out in Wuhan in 2019-20, but the research is done in 2022. What is the connection? Please define the seriousness of the pandemic situation and the measures used in Wuhan at the time of the investigation.

Page 2, 1st whole paragraph:

-        is the reference [3] correct here?

Page 2, 1st whole paragraph:

“health anxiety as a psychological problem is the cause of job burnout in medical staff [7].”

-        is the reference [7] correct here?

Page 2, 2nd paragraph

“However, this study found”

-        obviously “this study” refers to [12]. Please change into “The study” (omit however) and add a reference at the end of the paragraph to clarify what belongs into the reference

Page 3, 1st paragraph

“while the inhibitory effect of negative situations on burnout has been neglected.”

-        I do not understand how negative situations could have inhibitory effect on burnout.

Page 3, 1st paragraph

“In recent years, scholars have called for a focus on the negative factors of burnout, which have a stronger impact on individuals than positive factors due to the “positive- negative asymmetry”

-        Please add reference to studies.

“Thus, this paper will investigate the mechanism of health anxiety that influences job burnout from a demand perspective”

-        Previously there was also the role of self-efficacy.

2.2.1

- Here you define health anxiety. The definition should be in the introduction.

- Self-efficacy is not included in the title even though it should be one determinant. I suggest combining 2.2.1 and 2.2.2 for clarity.

- Please add the measure Pandey et al [22] used

“Job burnout is a state of work-related psychological exhaustion and an increased risk for somatic diseases was highly associated with the exhaustion dimensions of job burnout.”

-        Please add reference.

Page 4; Hypotheses:

you have written about sports and job burnout and chronic illnesses, but not much about health anxiety in regard to these, hence, the health anxiety appears in the hypotheses come rather suddenly. Please keep the “red line” in the text.

2.2.2

-  I suggest moving the definition of self-efficacy into introduction and combining 2.2.1 and 2.2.2 into one chapter.

- I have difficulties in following the role of sports to self-efficacy; the chapter discusses the role of sports for psychological well-being and physiological health, anxiety and depression, and mood, emotions and resilience, Later I read that self-efficacy has an influence on anxiety and depression, mental health and well-being etc. But this does not help me understand the connection between sprots and self-efficacy. Figure 1 is a nice representation of the study (but the text does not support it).

Page 5, top

Omit: “This section may be divided by subheadings. It should provide a concise and precise description of the experimental results, their interpretation, as well as the experimental conclusions that can be drawn.”

Demographics: Please add the share of females and males in the medical staff in general; is the share you had similar to that of the population? Tell the same concerning the age. Were those who did not respond males of females? Young or old? What does it mean to your study that there was 1/3 of males and 2/3 of females?

You write “Out of 500 distributed questionnaires, 56 uncompleted and inaccurate questionnaires were excluded, providing 444 valid responses, with an 88.8% effective response rate”. Do you really mean that you got back 100% of the questionnaires but some were uncompleted or inaccurate? It is difficult to believe such a participation rate in a voluntary study even though the participants earned a gift.

Please also define the snowball sampling already here.

Page 6; measures

 “All constructs were measured using a five-point Likert scale, in which the responses ranged from 1 (i.e., completely disagree) to 5 (i.e., completely agree).”

-        At least the MBI-GS I have used has a seven-point rating scale ranging from 1 (never) to 7 (daily).

Some scales measured negative experiences (health anxiety) and some positive (sports involvement, self-efficacy). The effects can, thus, be negative or positive, and this will affect the final model. Should health anxiety scale be inverted so that high scores indicate low health anxiety?

3.3

"Finally, this study used the Bootstrap method proposed by MacKinnon et al. [42] to test the mediating effect of health anxiety and self-efficacy with a sample size of 5,000 and a 95% confidence interval.”

-        Why do you need this analysis? You had a sample of 500 and a very high respondent rate, why not report that without any bootstrap? Further, at the end of the article you state that you “adopted the snowball sampling method” which might lead into a biased results (which affects in bootstrap as well).

4. Results

“Second, all the CR values were higher than 0.7, as suggested by Joseph F. et al. [44].”

-        Should this be Hair et al? (at least [44] is that.

“implying an acceptable convergent validity. Therefore, all variables have high convergent validity.”

-        Acceptable or high?

Table 2: You do not tell what do the loadings stand for. You mention Harmon’s [should be Harman’s] single factor test but without explanation it is impossible to understand the results. You state that  “The maximum factorial variance explained was 30.13%; thus, there was no common method bias in this study.“ Why does a certain level of explained variance lead to absence of common method bias?  Please add some knowledge about the test.

Table 3: there is no bold print in table 3. Further, what does the ‘Mean’ stand for and how was it calculated and what were the min and max; what is ‘MD’ (in text you mention standard deviation, which is SD), and what are the ‘Person’s corrections of contracts’? Where do you show the correlations between the variables (actually, measures here) for the reader to see that they are less than the square root of the AVE?

4.2

“The structural modeling results indicated that the final model has a better fit: χ2/df = 2.146, GFI = 0.951, NFI = 0.942, CFI = 0.968, TLI = 0.959, RMSEA = 0.051,”

-        Please explain the abbreviations (GFI, NFI, CFI, TLI, RMSEA); “has a better fit”: better than???

“Significant correlations were found between the independent various variables [?], the mediators, and the dependent various variables [?], which provided preliminary support for the verification of six hypotheses.”

-        Where do you report these correlations?

Please elaborate the  actual meaning of the β-values here; they vary between -.149 to .592;; are they low or high (please notice that a result can be statistically significant even though it shows that variable a has very small or even effect on variable b); please elaborate what the negative sign indicates (more sprots involvement has a negative effect on health anxiety, and more self-efficacy has a negative effect on job burnout), and how those negative effects affect the model.

4.3

“The conceptual model suggested that sports involvement has an indirect impact on job burnout through the mediation of health anxiety and self-efficacy.”

-        In the bootstrap analysis we can see the same negative effects as in the main analysis. And we can also see that some effects are high and some are low (even though they are all statistically significant). This analysis is redundant (and because of the snowball sampling, possibly even biased), in my opinion. If kept, the effects should be elaborated in the text.

Discussion

“the two mediating variables of health anxiety and self-efficacy explained 18.3% of the variance in job burnout.”

-        It is clear that the total variance explained was low, since there were both positive and negative effects. If all scales were turned into the same direction, the effect would be higher.

-        Please also state the 18.3% also in the results section.

“Second, this paper is the first attempt to explain the effects of sports involvement on health anxiety, self-efficacy, and job burnout using JD-R theory.”

-        This is difficult to follow since section 2 is not clear in terms of JR-D theory. For example, you refer to Bakker & Demerouti only in 5.2, even though one could expect that already in chapter 2.

5.2

“Long-term burnout not only depletes the physical and mental health of medical staff and increases their turnover rate but also reduces the quality of the healthcare services provided”

-        is this only true for health-care staff?

Conclusions

The conclusions section mostly double the findings and comments already written in Discussion. Additionally, it is a bit difficult to comment on the conclusions because of the many aspects that should be elaborated in the article.

References:

You have  Demerouti, E.; Bakker, A. B.; Nachreiner, F.; Schaufeli, W. B. The job demands-resources model of burnout. J. Appl. Psychol. 2001, 86 (3), 499-512. Twice in the list (13 and 17)

Author Response

(The authors gave the same response as above.)

Round 2

Reviewer 2 Report

This is more clear than the earlier version, thank you for the revision. Especially, the introduction leads into the research questions, the hypotheses are clear now and so is the text in 4.3.

Yet, there is still some work to be done. Please notice that I do not want everything explained to me in the cover letter; instead, I would like to see text clarifying the issues in the article. If I do not understand, there might be other similar readers as well.

1)     some issues are repeated in the introduction; e.g. all the following sentences are in the 2nd paragraph of the introduction. Please clarify this.

This led to increased anxiety and the risk of personal harm”

Health anxiety and high workload can lead to psychological distress and job burnout among medical staff [6],”

“and the psychological distress of health anxiety can lead to psychological disorders,”

“addition, health anxiety as a psychological problem is the cause of job burnout in medical staff.”

2)     about choosing Wuhan: The authors state in their cover letter that they have addressed my comment about why Wuhan, but the article text has not been changed (page 2): “The researchers chose Wuhan medical staff because Wuhan is the area where the COVID-19 pandemic broke out and was the first city to implement the lock-down policy.” You have added an explanation in 3.1, but it is too late there (and tells about participants, not the environment). Further, if Wuhan “won the battle against the virus in 2020”, it is difficult for the reader to understand how much extra work the Covid patients caused to the medical staff at the time of the study in April-May 2022. This might be irrelevant for the study per se; the role of sports involvement can be studied regardless of the pandemic. But it is still important for the reader to understand how the study relates to Covid-19. Can you skip the COVID-thing entirely?

3)     Chapter 3 (Methods): I asked about the possible bias caused by the snowball sampling, but you did not address this in the article. The cover letter tells even about a group activity organized for those who responded (if I understood correctly). Snowball can be a risk for bias, and group activities might lead to responses that are given to please the researchers. Further, I know what bootstrapping is. Since it uses the original sample to create simulated datasets, it is affected by the original sample and all possible biases included in that. For these reasons the issues should be addressed to in the article.

I also asked about the share of females and males in the medical staff in general, but you report that from the sample (actually twice in chapter 3.1).

In chapter 4 you still have "Loadings" on table 2 without any explanation in the text that they stand for Cronbach’s alpha (as they do according to the cover letter response); this is still more difficult for the reader to understand since the text states that “Table 2 exhibits the details of factor loadings, Cronbach’s alpha…” and these are separate both in the text and on the table. Please elaborate this in the text. Please remember that all readers might not know the method you use and you really should explain these details in the article text. The same is the case with abbreviations (GFI etc.) and the CMV + Harman’s test (by the way, I still believe it is Harman’s). The list of correlations are still missing and Table 3 lists “Person’s corrections of contracts” but in the cover letter the authors describe Pearson correlation coefficients. I was confused during the first round but now I am totally lost. Please explain these shortly in the text and make text and tables clear for the reader to follow.

4)     Re the explanatory power of the model, the total of 18.3% is relatively low. It might be because of the low coefficients between sports involvement and health anxiety (Please correct healthy  health in figure 2 and elsewhere); between self-efficacy and job burnout and between health anxiety and job burnout (the absolutes of these are lower than .4, some even lower than .2, indicating low or very low connection). As stated in my earlier review, statistically significant does not mean that there is a strong connection. I really wonder if a standardized coefficient .149 can be said to support any hypothesis. Please write about this in the text. The same continues in 4.3; if there is an effect 0.071 or 0.075, what do they mean? There is a small effect? Hardly any effect? A further detail concerns table 4: the text and Figure 2 calls the effects ‘standardized coefficients’ but Table 4 names them ‘point estimates’. Both might be correct but using two names for the same statistic makes it more difficult for the reader to follow.

5)     As stated, it is a bit difficult to follow the report and understand the strength of the effects. Hence, the sentences in 5.2. are problematic. First, “This study confirms the mechanism of the effect of sports involvement on job burnout in medical staff; in addition, we should also see the contribution of sports involvement to the individual’s physiological and psychological aspects.” There is some effect, yes, but below you define the impact as significant: “Considering the significant impact of sports involvement on health anxiety, self-efficacy, and job burnout, this study makes the following suggestions for medical staff.” As stated, an effect can be statistically significant, indicating that it is real, not caused by chance, but it might anyway be low, even non-existing.

Minor details:

p. 2:

However, this study found the influence of physiological factors”

please define what “this study” refers to

Sports involvement is indirectly associated with job burnout, mediated by self-efficacy and health anxiety.”

-        does this added sentence tell about your results? If so, it should not be in the introduction. If it refers to earlier study, please refer to that study. If it is a hypothesis-like sentence, please define that.

chapter 2.2  explains the connection between stress, health anxiety, job burnout, sports and self-efficacy. Since these issues appear in this order, I suggest using this order in the title as well. (please also edit healthy anxiety  health anxiety)

Chapter 2.2. Please add one sentence between the text and the hypotheses telling that according to earlier study etc. the hypotheses are….

Table 4. The direct effects and total effects seem to be the same, please omit redundant information or add correct effects (if not the same)

Chapter 6

This study has certain limitations. First, this survey respondents were more female medical staff than male, which limited the representativeness of the sample to some extent.” Actually the reader does not know if the sample was “limited in representativeness” since there is nothing in the article about the gender differences in the medical staff in China (nor Wuhan area); you only report that of the sample. Please elaborate.

Author Response

Dear reviewer,
We sincerely thank you for examining our manuscript and providing helpful comments to guide our revision. We have tried our best to revise the manuscript according to your kind and construction comments and suggestions. Please find the following detailed responses to your comments and suggestions. We would like to thank the referee again for taking the time to review our manuscript.
Kind regards,
Xiuyu Chen
